# Collaborative Use of a Shared System Interface: The Role of User Gaze—Gaze Convergence Index Based on Synchronous Dual-Eyetracking

**Armel Quentin Tchanou \*** , **Pierre-Majorique Léger \*, Jared Boasen \*, Sylvain Senecal, Jad Adam Taher and Marc Fredette**

Tech3Lab, HEC Montréal, Montréal, QC H3T 2A7, Canada; sylvain.senecal@hec.ca (S.S.); jad.taher@hec.ca (J.A.T.); marc.fredette@hec.ca (M.F.)

**\*** Correspondence: armel-quentin.tchanou@hec.ca (A.Q.T.); pierre-majorique.leger@hec.ca (P.-M.L.); jared.boasen@hec.ca (J.B.)



**Featured Application: Synchronous gaze recording facilitates dual eye-tracking data processing and permits real-time assessment of dyad-level (of group-level) constructs. The proposed dyad gaze convergence index may help empirically investigate dyad convergence antecedents and consequences during collaborative use of information technologies. Synchronous real-time gaze convergence visual cues may improve collaboration and dyad performance.**

**Abstract:** Gaze convergence of multiuser eye movements during simultaneous collaborative use of a shared system interface has been proposed as an important albeit sparsely explored construct in human-computer interaction literature. Here, we propose a novel index for measuring the gaze convergence of user dyads and address its validity through two consecutive eye-tracking studies. Eye-tracking data of user dyads were synchronously recorded while they simultaneously performed tasks on shared system interfaces. Results indicate the validity of the proposed gaze convergence index for measuring the gaze convergence of dyads. Moreover, as expected, our gaze convergence index was positively associated with dyad task performance and negatively associated with dyad cognitive load. These results suggest the utility of (theoretical or practical) applications such as synchronized gaze convergence displays in diverse settings. Further research perspectives, particularly into the construct's nomological network, are warranted.

**Keywords:** gaze convergence; eye-tracking; synchronous dual-gaze recording; dual eye-tracking; collaborative use; shared interface

## 1. Introduction

User gaze during interactions with information technologies (IT) has been the object of increasing interest in management research. Multiple research works in the field of human computer interaction (HCI) have investigated user's eye movements during system use (e.g., Cyr et al. [1]; Belenky et al. [2] ). In this regard, constructs related to IT user's gaze have been related with different information system (IS) -related constructs, including interest, attention, cognitive load, and confusion [3–6]. Nevertheless, these studies have essentially focused on single-user settings. Consequently, even in the context of collaborative use, IT user's gaze indicators have been essentially recorded separately for each participant [7,8]. Indeed, studies investigating simultaneous collaborative use of a shared interface are scant in the extant literature. Thus, the true role of users' gaze in this context of IT collaborative use is not well understood.

Investigating gaze during collaborative use of shared system interfaces is important for several reasons. Firstly, gaze convergence of collectives of users, that is, the act of looking at the same location on a system interface, has been suggested to be important for group learning (e.g., Belenky et al. [2]) and group work (e.g., Kwok et al. [9]). Secondly, although most systems are designed with a single user in mind, they are frequently used in multiuser settings. Examples include individual productivity tools such as diagramming application [10] and e-commerce platforms [11]. To illustrate further, a recent study revealed that 53% of online household purchases are operated by multiple users shopping online together [12]. These multiuser settings involve users gazing at each other or at visual objects of interest in the system interface. Finally, during collaborative interactions, users may relate to IT artefacts visualized in the system interface. Indeed, users will navigate a given interface according to their own mental models, which can also be modified during navigation. Mental models are mental elaborations of user's understanding of knowledge and relationships between concepts or systems [13]. It is thus desirable that collaborating users have similar mental models regarding the system to facilitate collaboration. It has been suggested that mental model construction may be induced by eye movements, through which individuals learn spatial structure of visual elements (e.g., Eitel et al. [14]; Schnotz et al. [15]). Hence, collaboration during joint interactions with system interface may be facilitated when user gazes are convergent.

We investigated the eye movements of dyads of users during their collaborative use of a shared system interface. More precisely, we measured the extent to which the user dyads exhibited gaze convergence by looking at the same locations on the screen during system use. Moreover, we investigated the influence of gaze convergence on information system (IS) use-related constructs including cognitive states and task performance, raising the following research questions (RQ).

RQ1: How can the degree of gaze convergence of a dyad collaborating simultaneously on a shared system interface be measured?

RQ2: To what extent does gaze convergence relate to dyad cognitive states and dyad performance?

The present study sought to answer these questions methodologically and theoretically. Firstly, we empirically illustrate feasibility of simultaneously and synchronously recording eye-tracking data of user dyads, a technique still embryonic in the HCI literature. Secondly, we propose a novel index for measuring gaze convergence (GC) of user dyads, a construct still scant in the literature. Finally, we examine the role of gaze convergence in relationship with system use-related constructs, one of the very few such initiatives in the literature.

In the remainder of this paper, we test the dyad GC index and examine its validity through two consecutive studies. In the first study, we examine GC contruct validy. In the second study, we develop and examine a model of GC to assess predictive validity of the proposed gaze convergence index. As expected, the dyad GC index clearly distinguished between gaze convergence and gaze divergence, and gaze convergence was found to be positively associated with dyad performance and negatively associated with dyad cognitive load. Concluding discussions follow, including research perspectives.

## 2. Theoretical Development

### 2.1. Gaze Convergence

A literature review was conducted to assess interest in the GC construct. The keywords "gaze convergence" or "shared gaze" or "scanpath comparison" were used. The search was performed mainly through some of the most prominent databases, including Web of Science, ABI/INFORM, Wiley Online Library, and ProQuest. We found no study investigating GC in the context of collaborative system use. In past research, the concept of GC has been mainly investigated in terms of mutual GC, often referring to people looking at each other (e.g., Thepsoonthorn et al. [16]). These works are mostly focused on face-to-face communication between avatars (e.g., Wang et al. [17]), between robots and humans (e.g., Thepsoonthorn et al. [18]), and between humans (e.g., Thepsoonthorn et al. [19];

Thepsoonthorn et al. [16]). Clearly, HCI research on GC during collaborative system use is still embryonic, and the question as to how to measure it remains unexplored.

In the present research, we examine two types of GC: system-oriented GC (SOGC) and mutual GC (MGC). In this study, we define SOGC as the extent to which a dyad of users look around the same locations on a shared system interface during collaborative system use. This definition considers that the two users are exposed to the same system interface layout, through the same monitor or through separate display devices with or without same dimensions. This kind of setting can be achieved via single display groupware systems, which allow coworkers to collaborate using desktop computers and mobile devices displaying the same system interface [20]. Meanwhile, MGC is defined in this research as the extent to which a dyad of users look at each other while collaborating on a shared system interface. This construct has a distinct content domain from the SOGC construct, since it focuses on dyad gazes directed to locations completely out of the system interface and monitor.

### 2.2. Eye-Tracking Technology

The use of eye-tracking (or oculometry) is emerging and informs research in IS and HCI [21]. This trend is illustrated by the fact that for the past decade, eye-tracking has been the most used neurophysiological tool in NeuroIS research [22]. Eye-tracking is a technique permitting measurement of eye movement and gaze location, providing a researcher, at any point of time, with information about what stimulus is being processed by a user [6]. To capture a user's eye movements, eye-tracker systems target physiological characteristics of the eye with infrared technology along with high-resolution cameras [6]. This technique uses image processing software to capture two eye features [4]: the corneal reflection appearing as small bright glint on the eye pupil, and the center of the pupil. The analysis software finds the position of the user's gaze on the screen based on the relative position of the pupil center and the glint, along with trigonometric computation [4].

Two important eye-tracking elements are saccades and fixation. Saccades are the short duration eye movements (ranging between 30 ms and 80 ms duration) with no information processing [6,23]. Fixations are short stops between saccades [4] generally lasting a minimum of 50 ms (e.g., for text processing) or 150 ms (e.g., for image processing). Fixations are usually analyzed at specific area of interest (AOI) defined by the researcher [6,23]. There are five main eye-tracking measures that are usually employed: fixation duration (amount of time a point is fixated by the user), fixation frequency (or fixation counts: number of times a point is fixated), time to first fixation (the time it has taken the user to gaze inside an AOI), visit count (number of times a viewer's gaze entered an AOI), and total visit duration (length of time a user gaze remained in an AOI).

Additionally, there are two main gaze representations used with eye-trackers: fixation patterns and gaze heatmaps. Fixation patterns are two-dimensional plots of the fixation points for a given user. Gaze heatmaps are heatmaps made of the aggregation of user fixation patterns, with fixation intensities being represented using gradients of a discrete set of colors (red for high, yellow for moderate, and green for low intensity) [24].

### 2.3. Synchronous Dual Gaze Recording

As the ubiquity of eye-tracking technology has increased over the past decade, so too has its affordability [7]. Today, there is far more potential to employ eye-tracking in studies regarding collaborative IT use. Correspondingly, eye-tracking experiments with multiple participants have become more common [7,25]. However, a major limitation of prior multiuser eye-tracking studies is that the recordings have essentially been done separately for each participant and have commonly not been synchronized, making eye-tracking data analysis tedious and imprecise. Indeed, not synchronizing the recording computers means that their clocks may not be linked, permitting an artificial temporal mismatch between concurrent actions between the participants. Although realignment of the data can be achieved by using timestamps, the procedure is prone to inaccuracies or errors [7], and it is time-consuming and complicated. However, synchronous recording of eye gaze of multiple participants

has started gaining some attention, notwithstanding the use of low-cost, low-accuracy eye-tracking devices [25].

To address the above concerns, the present study implemented synchronous dual gaze recording through a high-accuracy eye-tracking setup. All data recording computers for user dyads were synchronized during simultaneous collaborative use of a shared system interface. We used a dual eye-tracking method involving participants sitting each in front of a separate display. Compared to setups in which one display is shared for all participants on the same device, our method provides a higher accuracy of the eye-tracking data, since it is as precise as the eye-tracking method with one participant [25]. An important advantage of synchronized dual eye-tracking is that it brings order to the gaze data files and eases later data analysis, allowing to track exact time and order in which the gaze data were collected [7]. For details, please see the Methodology section.

## 3. Hypothesis Development

### 3.1. Gaze Convergence Index

We built a GC index based on the gaze of each dyad member (see Figure 1). As mentioned earlier, we conceptualize GC as comprising two formative dimensions, namely SOGC and MGC. For the sake of parsimony, and as an initial effort, the present paper examines the GC index through the SOGC dimension. Hence, in the remainder of this paper, GC is defined as the extent to which a user dyad look at the same locations in a system interface. We propose two reflective dimensions of SOGC: real-time gaze distance (RTGD) and overall fixation distance (OFD). RTGD is defined as the distance between the gaze fixation point of each dyad member at any given time. Thus, when dyad members look near the same location on the screen at any given point in time, RTGD will be small and GC will be high. Meanwhile, OFD is defined as the extent to which dyad members have overall looked at the same locations on the screen. On this basis, GC will be considered high for the duration of a task if dyad members have generally looked at the same locations with similar intensities.

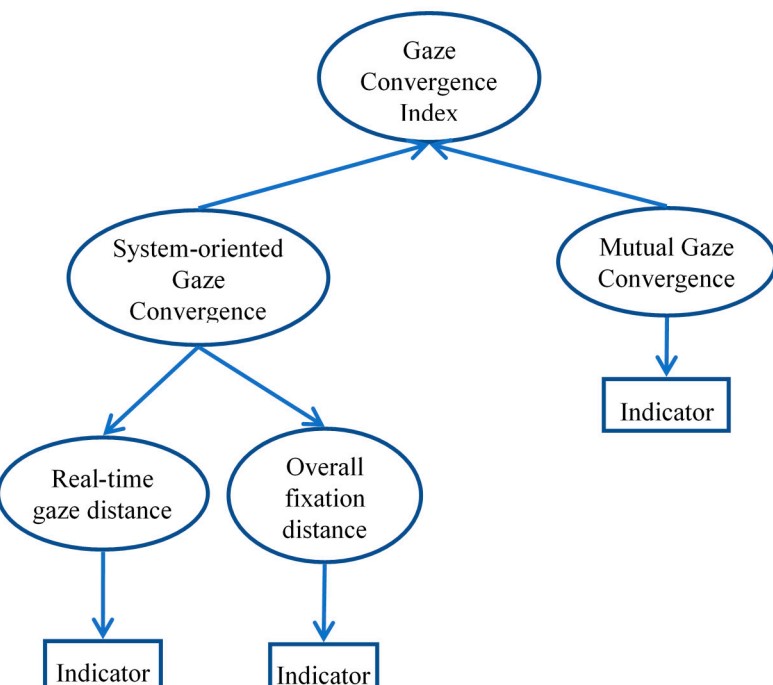

**Figure 1.** Gaze convergence index construct. Note. The oval forms represent constructs and the rectangles represent direct measure of the associated subconstruct.

These definitions hold true irrespective of the display device as long as the dyad members interact with the same interface with the same visual layout. The proposed GC index is depicted in Figure 1.

### 3.2. Dyad Gaze Convergence and Its Impact

In order to assess the predictive validity of the proposed GC index, we developed a model to examine salient associations with GC in the context of a user dyad collaborating simultaneously on a shared system interface. As gaze convergence is the focus of the present paper, our model is developed in the context of tasks performed jointly using a shared system interface, with the users having the same objective and focus. Figure 2 depicts the research model.

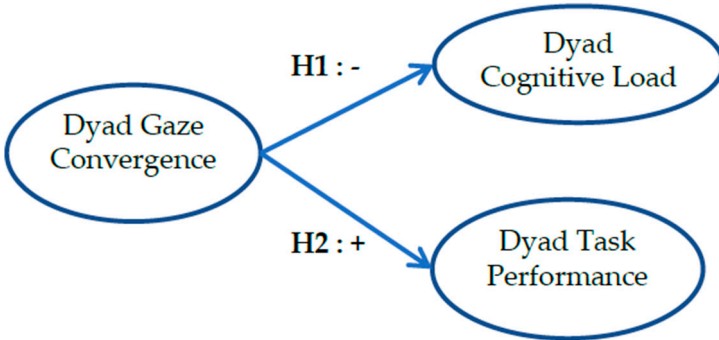

**Figure 2.** Model for predictive validity of dyad gaze convergence.

User cognition has been frequently investigated in the IS field and has been referred to as what occupies individual's mind and that s/he is aware of or not while using a system [26,27]. Cognitive load has been a prominent construct of cognition studied in the HCI and IS disciplines, considered an indicator of efficient use of a system [28,29]. System user cognitive load has been referred to as the extent to which mental resources are employed by users to encode, activate, store, and manipulate data or information as they use a system [29,30]. The present study adopts this conceptual definition in the context of user dyads collaborating simultaneously on a shared system interface. Cognitive load at dyad level results not only from each individual mind alone but also from interactions between team members, that is, it also happens socially [31]. Users process information not only from the system interface but also the other dyad member. In the context of multiple users, some level of individual effort is needed in order to reach common objectives. Moreover, individual efforts have to be coordinated and aligned to ease collective task achievement [32]. Yet, the effort required for coordination could conceivably be eased when the dyad members share mental models [33] of the system interface they use together. A user's mental model within any time range may develop from how and where he or she looks at the system interface. Thus, mental model is a mechanism through which the user generates descriptions of system purpose and visual form. This permits comprehension of system functions and observed system states, and prediction of future system states [33]. Consequently, when dyad members working together in real-time on a shared visual interface do not look at the same regions in the shared interface, they may not share the same mental model at specific moments, consequently requiring more cognitive resources to communicate and coordinate actions. Thus, we could hypothesize that the more users look at the same regions on the visual interface the less cognitive resources they will need to collaborate. Moreover, two collaborating users looking at the same locations are likely to perform better, as they would be able to better coordinate their interaction. Past research suggests that looking at the same visual objects in a system interface reduces miscoordination incidents, hence improving dyad coordination (e.g., Thepsoonthorn et al. [16]; Zhu et al. [34]). Yet, coordinated efforts facilitate the achievement of collective task, increasing team performance [32,35,36]. Hence, we make the following hypotheses (H).

**H1:** *Dyad GC will be negatively associated with dyad cognitive load.*

**H2:** *Dyad GC will be positively associated with dyad task performance.*

## 4. Methodology

To validate our proposed GC index and investigate its hypothesized associations, we conducted two consecutive experimental studies, which were approved by the ethical committee of our institution, the Comité d′Étique de la Recherche (ethical approval code: 2020-3645). The two studies involved synchronous dual eye-tracking recording. In other words, eye movements of user dyads were recorded with real-time synchronization of clocks on two gaze data recording computers. Informed consent was obtained from all dyad members prior to participation.

The first study assessed the content validity of the proposed GC index. Content validity refers to the degree to which a construct operationalization is representative of the content domain (i.e., the substance, the matter, or the topic) of the construct [37–40]. 37-40 Simply put, we ascertained to what extent our GC operationalization reflected the GC of user dyads. To that end, GC was experimentally manipulated by having user dyads perform a task in two conditions: gaze convergence (referred to as the convergence condition) and gaze divergence (referred to as the divergence condition). Hence, the purpose of this study was to assess the extent to which our dyad GC index is able to distinguish between convergent and divergent dyad gaze behaviors.

The second study explored the predictive validity of the proposed GC index, that is, the extent to which the operationalization of the construct was able to correlate with or predict endogenous constructs it is theoretically expected to correlate with or predict [40]. Specifically, this study tested whether dyad GC would predict the hypothesized dependent variables, namely, dyad cognitive load and dyad task performance. User dyads engaged in collaborative tasks using the same interface on different computer monitors, with only one of the two users controlling input devices (see Figure 3).

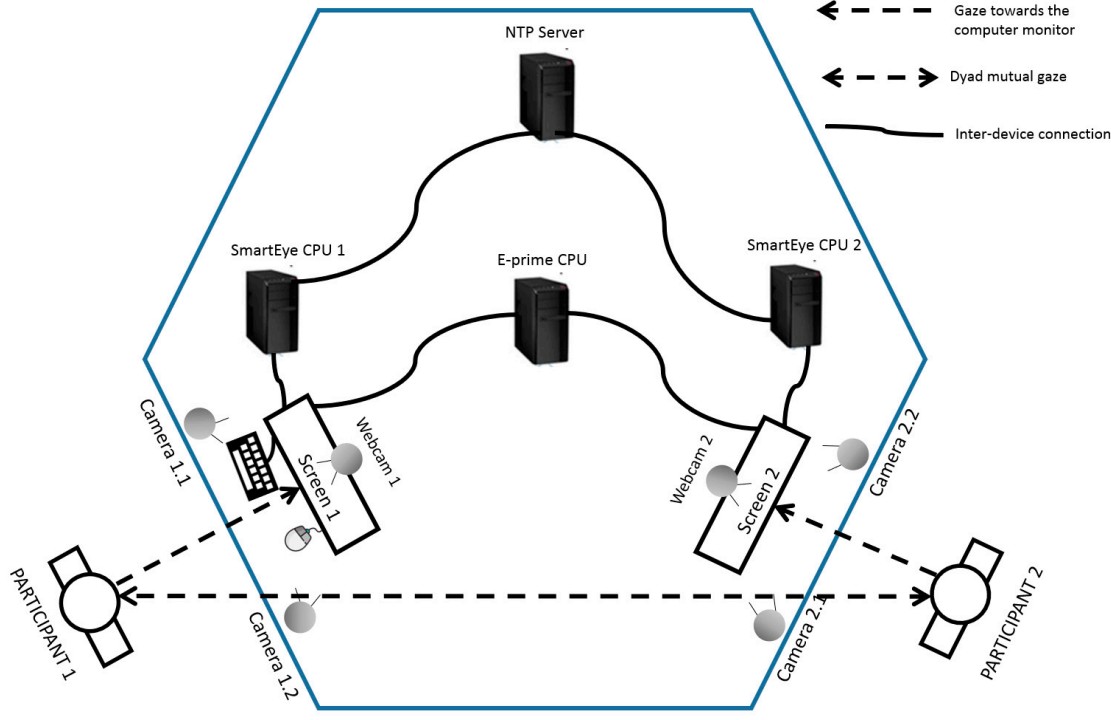

**Figure 3.** Eye-tracking experimental setup.

This setup reflects a specific real-world setting in which users work together on the same task, on a shared system interface, to jointly perform a task. Examples include the following: two workers sharing their screen (e.g., using a screen sharing software) to work together from same or separate



offices or rooms but communicating directly of through the phone; or two IT network professionals jointly working on a shared interface with separate computer monitors in a technical equipment room, as each worker needs to stay at different locations to monitor different equipments. This setup is also common among couples who shop online together remotely, as revealed by a forthcoming paper about the state of the art on multiuser human-computer interactioin settings, providing preliminary evidence specific to settings in which couples shop online together [41]. In that survey study, more than 70% of couples shopping online together using separate screens reported sharing the same software window during the activity.

## 4.1. Material and Apparatus

We used SmartEye Pro (SmartEye, Gothenburg, Sweden) eye-tracking system, which enables a non-invasive temporally synchronous collection of gaze and pupil data of multiple participants. This system employs two cameras per participant for large field of view and permits measurement of a head orientation and gaze direction in 3D, including the ability to discriminate between gaze directed at the interface or at the other user in the dyad. SmartEye Pro also permits measurement of pupil diameter in real-time during the task, and it provides a great gaze accuracy (around 0.5 degrees [42] for a 30 cm eye distance—a 1 degree gaze accuracy is considered high in the eye-tracking literature [25,43–45]). It was configured with a sampling rate of 60 Hz. To synchronize computer clocks at all times during the studies, we configured a Network Time Protocol (NTP) Server. NTP protocol is designed to synchronize several computers' clocks across variable-delay networks, with an accuracy below one millisecond between network devices [7].

The experimental stimuli were developed and administered using E-Prime 3.0 software (Psychology Software Tools, Sharpsburg, MD, USA). The software ran on a computer with a clock perfectly synchronized with SmartEye computers through the NTP server. A crucial benefit of E-Prime is that it is able to provide a rich set of time stamps for every event or display, allowing for direct matching with our resulting eye movement data. Moreover, E-prime permitted synchronized acquisition of questionnaire data during the experimental tasks. The stimuli were run on two identical computer monitors connected to the same computer central processing unit (CPU) to permit shared display.

In addition to the two SmartEye cameras, a video camera was fixed on the top of each user's computer monitor. The audio and video of each user's face, which were sampled by these cameras during the two experimental tasks were recorded with Media Recorder software (Noldus, Wageningen, The Netherlands), thereby permitting future assessment of behaviors such as head orientation and characteristics of auditory communication. Observer XT software (Noldus, Wageningen, The Netherlands) synchronized Media Recorder recordings to our eye-tracking recording through time stamps that were accurately linked to the absolute time of the study as provided by the NTP server.

Figure 3 depicts the physical setup used for both studies.

## 4.2. Users

Our experimental sample comprised 8 dyads, or 16 users (5 females and 11 males), with an average age of 24.1-year-old and a standard deviation of 2.6-year-old, recruited through our institution's recruitment panel. To participate in our study, panel members had to be 18 years or older and could not have specific skin sensitivity or allergies, a history of epilepsy, neurological or any other health related disorder, nor use a cardiac pacemaker. The recruitment was done with no requirements over whether participants knew one another, and dyads were formed randomly.

## 4.3. Experimental Procedure

One user dyad was scheduled for every experimental session (both Study 1 and Study 2). After welcoming users and ensuring no exclusion criteria were met, two research assistants asked the users to sit comfortably at their respective computer desks and briefed them about the study's purpose. User dyads sat in a configuration allowing them to gaze at each other and communicate

during the tasks. Only one dyad member had access to the computer's input devices (mouse and keyboard) during every experimental task: as this study's hypotheses are not specific to any input control setting, giving that control to only one of the two dyad members allowed for more simplicity. It helped rule out possible events of no interest in the present study such as input control switching or negotiation. The research assistants proceeded to calibrate the SmartEye system for each user, based on the SmartEye system manual [42]. After all experimental tools were ready, the E-Prime executable file was run in full-screen mode throughout the duration of the experimental session. At the beginning of the first part of the experimental session (Study 1), participants were instructed to close their eyes and breathe for thirty seconds, then while each dyad member looked at an initial blank screen display, baseline eye parameters were recorded through SmartEye. The same process was followed at the beginning of the second part of the experimental session (Study 2), but the baseline data of the first part were used for the whole experimental session. SmartEye data were recorded throughout the entire session as well.

### 4.4. Study 1 Experimental Design

The experiment was a factorial design with repeated measures in two conditions: convergence and divergence. In both conditions, dyads were exposed to the same stimuli. The stimuli were a small blue and a small red solid colored circle moving in opposite directions along a single track of twelve fixed display positions. The two circles' displays were made to never overlap. Both circles displayed for five seconds and then immediately moved to their respective following positions. Thus, user gaze was expected to move steadily along in a stepwise fashion along each of the twelve circle positions.

In the convergence condition, dyads were asked to stare at the blue moving circle at all times. In the divergence condition, one dyad member was instructed to stare at the blue moving circle, while the other was instructed to stare at the red moving circle. Each user dyad was exposed to one trial of each condition, resulting in a total of sixteen trials and one hundred and ninety-two dyad eye. Figure 4 depicts the sequence of movements for the blue and the red circles in the experimental stimuli.

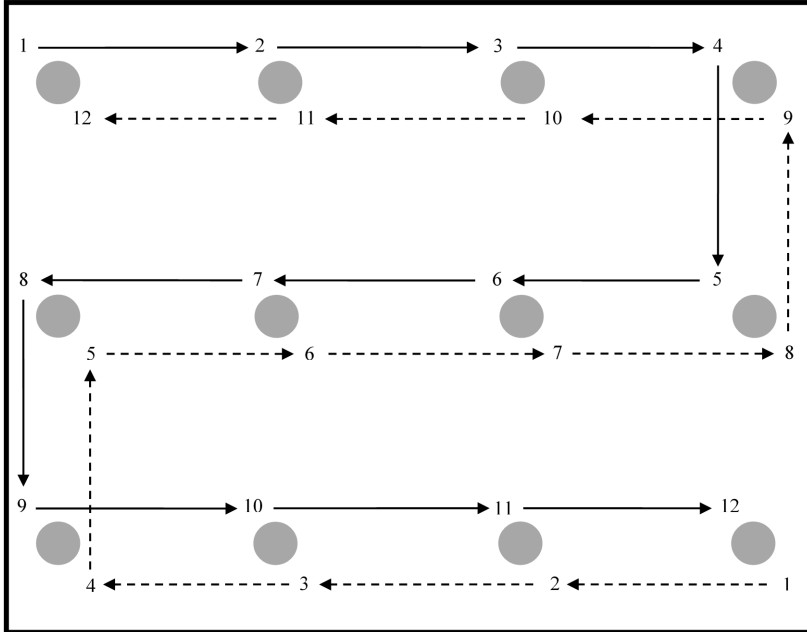

**Figure 4.** Experimental stimulus: circle movement sequences. The bold frame represents computer screen. The gray circles represent the twelve possible positions of either blue or red circle. Solid arrows represent blue circle's movement sequence. Dotted arrows represent red circle's movement sequence. Each ordered number sequence (from 1 to 12) depicts the direction of blue circle's course and the red circle's course, respectively.

### 4.5. Study 2 Experimental Design

In the second experiment, GC was not manipulated. Dyads had to perform a psychological task collaboratively, namely, a change blindness task. Change blindness is so named due to the psychological phenomenon where a difference between two nearly identical images becomes difficult to discern when the images are viewed after a small time delay [46]. In the present study, seven image pairs with a single subtle difference were used. The images in a given image pair were alternately displayed for one second with a one-second white blank display between them. This cycle of display was repeated for an unlimited amount of time. The cycle stopped when the dyad decided to answer the related question by pressing the "SPACE" key on the keyboard (they were instructed to do so only when the two dyad members were ready). Then a multiple-choice question displayed regarding what element of the image was changing. The question was phrased as follows: "What type of change did you notice?" There were a total of four answer options, including the option "We could not identify any change". For each image pair trial, dyad response time and final response choice were recorded. The experiment resulted in a total of fifty-six trials across all eight dyads. The change blindness task was chosen because it is a well-known psychological task that can be readily programmed in E-Prime software for experimental purpose. Moreover, the task was driven by dyad's common interest and objective in the visual interface. Besides, in order to further foster dyad collaboration, dyad members were told that they could collaborate during the task. The collaboration happened naturally, as participants were not limited in their movements.

### 4.6. Measures

As we are interested in where dyads look in the system interface, and since all E-Prime displays were in full-screen mode, we considered gaze location in the orthonormal plane represented by the computer screen. As suggested previously, dyad GC is high when the distance between gaze locations is low, and vice versa. We measured dyad GC in both studies as the opposite (i.e., multiplied by $-1$) average Euclidian distance between gaze locations of the two dyad members at all fixation points. The fixation data of both users were synchronous in time, allowing for direct comparison. Similar measures based on Euclidian distance but in context of asynchronous gaze recording have been reported to be valid measures [47–49].

In Study 1, in order to get a more comprehensive assessment of the distribution of distance between dyad gaze locations (see time series of distances in Figure 5), other statistics of Euclidian distance between dyad gazes were planned for analysis, including minimum, standard deviation, 1st quartile, 2nd quartile (i.e., the median), 3rd quartile, and maximum. These analyses would reveal whether the difference between convergent and divergent gaze behavior detected by our dyad GC index is supported throughout the gaze distance data distribution, from lower (lower GC) to higher (higher GC) data points.

In Study 2, two other constructs were measured: dyad task performance, and cognitive load. We operationalized dyad task performance to be associated with higher accuracy and shorter response time. Accuracy was scored according to the following: 1 point for incorrect responses, 2 for unknown answers, and 3 for correct responses. This grading was in line with the nature of the experimental task. As every image pair had a single subtle difference and dyads had unlimited time for double-checking, it was more expected that user dyads would either find the right difference or abandon the trial without finding the image difference. Hence, incorrect response was the most penalizing answer. The sum of points for all trials in Study 2 determined a dyad's Accuracy Score. Moreover, the total time taken by a dyad to analyze the stimuli for each trial before responding was deemed as the Trial Completion Time. Dyad task performance was measured as follows:

Task performance = {Accuracy Score}/{Trial Completion Time}

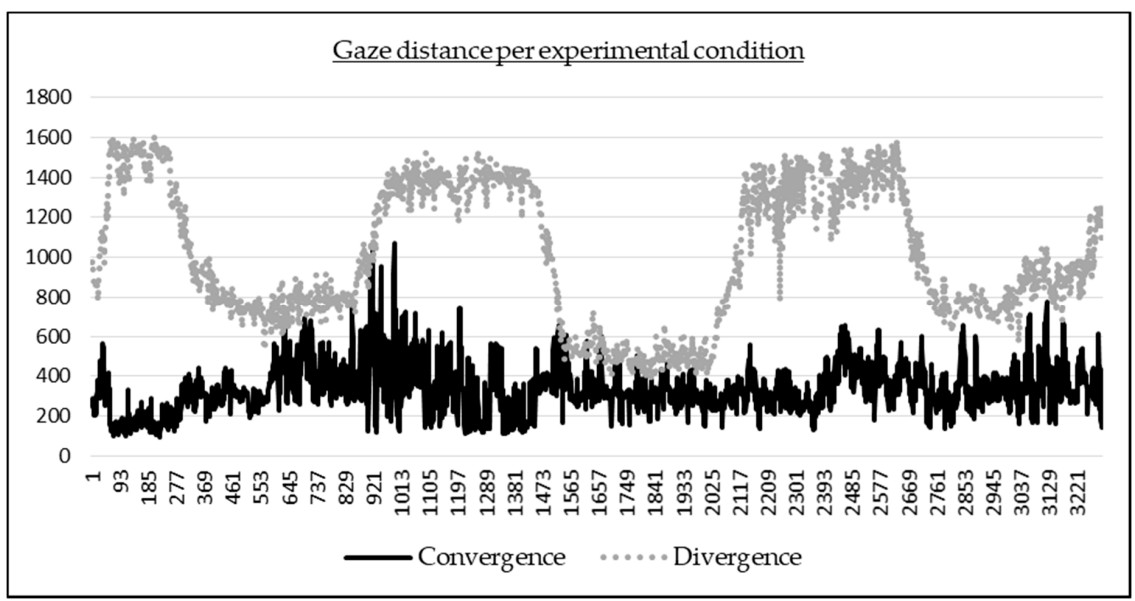

**Figure 5.** Average gaze distances within dyads in each experimental condition. The sinusoidal trend in the Divergence condition is consistent with the fact that the blue and the red circles alternatively moved closer to or farther from each other overtime.

Meanwhile, cognitive load was indexed via pupillary dilation, a common strategy for human-computer-interface studies in the IT literature [22,50,51]. Specifically, cognitive load for an individual user was determined as the increase (or decrease) in pupil diameter during the task, that is, pupil diameter recorded during the main experimental task minus baseline recorded before the start of the experiment. Dyad cognitive load was determined as the mean cognitive load between the two users in the dyad. This operationalization of dyad cognitive load is in line with past studies of team-level cognitive load (e.g., Gopikrishna [52]; Litchfield and Ball [53]; Lafond et al. [54]).

*4.7. Statistical Analyses*

In study 1, manipulating gaze convergence permitted to assess content validity of dyad GC index, that is, the extent to which our measure captures gaze convergence. To do so, we used Wilcoxon signed-rank test [55] with Statistical Analysis Software (SAS). This statistical method is particularly useful when the population is not assumed to be normally distributed and the sample size is small. Because our hypothesis was directional, we used one-tailed *p*-values.To get further insights related to patterns of task performance by dyads, we ran an ANOVA with a two-tailed *p*-value. Here we analyzed trial completion time to check for differences relative to the answer accuracy. This analysis would inform us whether dyads took different time to provide correct, wrong, or "unknown" answers. The significance level was set at $\alpha = 0.05$.

To test hypotheses H1 and H2 through the change blindness experiment (Study 2), we used a linear regression with random intercept model. Hence, we tested negative association between dyad GC and dyad cognitive load measured using pupillary dilatation as mentioned earlier. We also tested negative association between dyad GC and dyad task performance. Moreover, we used one-tailed *p*-values and significance level of $\alpha = 0.05$ for the same reasons as in the first experiment. Finally, it is important to recognize that the images used in the Change Blindness task were of lower luminosity than the blank white screen displayed immediately prior to and after each image. There is a possibility that as the time taken to complete the trial increases, the eyes become accustomed to the less luminous images, causing pupil diameter to gradually increase, and thereby causing an artificial increase in cognitive load. To control for this, we added trial completion time as a control variable in the statistical model.

## 5. Results

### 5.1. Study 1

Results revealed that GC index values were significantly higher in the convergence condition, compared to the divergence condition ($p = 0.004$; r = 0.19). This result is illustrated in Figure 5, which depicts a clear visual difference in GC between the two conditions, GC being the opposite of gaze distance as mentioned earlier. Moreover, a more detailed analysis showed significant differences in main statistics between the convergence and the divergence conditions. The convergence condition recorded the highest maximum of convergence between dyad members' gazes ($p = 0.004$; r = 0.31). Moreover, significantly higher GC values were recorded in the convergence condition for GC first quartile ($p = 0.004$; r = 0.05), second quartile ($p = 0.004$; r = 0.31), and third quartile ($p = 0.004$; r = −0.19). Besides, the convergence condition recorded significantly lower standard deviation ($p = 0.02$; r = 0.12). Clearly, the difference in GC between the two conditions was consistent throughout the data distribution. Thus, we are confident that the dyad GC index does measure gaze convergence (content validity is supported). Table 1 presents descriptive statistics for each treatment condition.

**Table 1.** Descriptive statistics, Study 1, per experimental condition.

| Variable | Mean | | StD | | 1st Qrtl | | 2nd Qrtl | | 3rd Qrtl | |
|---|---|---|---|---|---|---|---|---|---|---|
| | C | D | C | D | C | D | C | D | C | D |
| Dyad GC | −333.51 | −1013.53 | −153.55 | −107.67 | −468.28 | −1126.12 | −261.59 | −1007.32 | −223.20 | −916.55 |
| Min GC | −1832.14 | −1872.52 | −167.72 | −155.33 | −1992.26 | −2031.86 | −1831.51 | −1886.64 | −1661.59 | −1710.79 |
| Max GC | −5.39 | −36.61 | −4.12 | −17.86 | −7.30 | −46.05 | −4.49 | −34.88 | −2.03 | −25.26 |
| StD GC | −317.81 | −433.86 | −126.11 | −39.38 | −443.02 | −473.26 | −280.23 | −432.91 | −209.35 | −405.92 |
| 1st Qrtl GC | −127.96 | −653.06 | −39.72 | −81.14 | −166.37 | −722.63 | −135.78 | −664.00 | −93.14 | −570.49 |
| 2nd Qrtl GC | −224.02 | −931.74 | −94.87 | −157.93 | −293.32 | −1025.84 | −199.64 | −894.01 | −157.83 | −807.07 |
| 3rd Qrtl GC | −427.62 | −1428.92 | −277.82 | −125.41 | −631.48 | −1556.36 | −273.19 | −1404.40 | −247.67 | −1367.46 |

GC = Gaze convergence; StD = Standard Deviation; Qrtl = Quartile; C = Convergence condition; D = Divergence condition. The Variable column refers to descriptive statistics examined as independent variables. With a total of sixteen participants, our sample size was eight dyads in each treatment condition.

### 5.2. Study 2

Results of the linear regression with random intercept model, with trial completion time as a control variable, showed that dyads with higher GC during the task exhibited significantly less pupil dilatation ($p < 0.001$; t = −4.09), supporting hypothesis H1. Moreover, dyads with higher GC performed significantly better during the task ($p = 0.003$; t = 2.90), supporting hypothesis H2. Table 2 presents descriptive statistics, complemented with statistical results in Table 3 and Figure 6. Figure 7 depicts the plots for the two regression models. Finally, post-hoc regression with cognitive load as dependent variable and trial completion time as independent variable was non-statistically significant ($p = 0.810$; R2 = 0.001; F = 0.058).

To check for differences in dyad performance at trial level, we ran an analysis of variance (ANOVA). The test was significant at $\alpha = 0.05$ (F = 5.629; $p = 0.006$). Simple contrast analysis showed that dyad trial completion time was significantly lower for right answering than for "unknown" answering ($p = 0.005$; C.I. = [0.478; 2.626]) and than for wrong answering ($p = 0.043$; C.I. = [0.064; 3.628]).

**Table 2.** Descriptive statistics, Study 2.

| Variable | Mean | StD | 1st Qrtl | 2nd Qrtl | 3rd Qrtl |
|---|---|---|---|---|---|
| Dyad GC | −281.62 | −99.64 | −370.05 | −301.91 | −221.05 |
| Dyad TP | 2.05 | 0.66 | 1.75 | 1.99 | 2.16 |
| Dyad PD | 0.38 | 2.72 | −0.79 | 0.36 | 2.59 |

GC = Gaze convergence; TP = task performance; PD = pupil diameter; StD = Standard Deviation; Qrtl = Quartile. With a total of 16 participants, i.e., 8 dyads for a total of 56 trials.

**Table 3.** Statistical results, Study 2.

| DV | Effect Estimate | DF | t | *p*-Value |
|---|---|---|---|---|
| Dyad TP | 26.19 | 46 | 2.90 | 0.0029 |
| Dyad PD | −15.99 | 46 | −4.09 | 0.0001 |

DV = Dependent variable. TP = task performance; PD = pupil dilation; DF = Degrees of freedom.

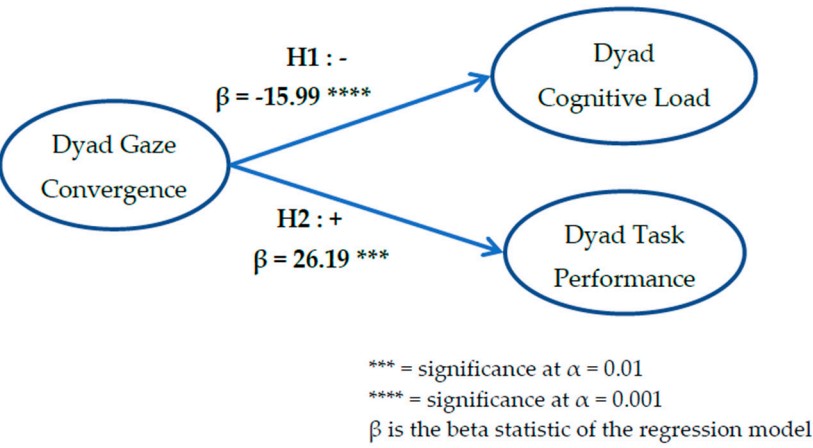

*** = significance at α = 0.01
**** = significance at α = 0.001
β is the beta statistic of the regression model

**Figure 6.** Results, Study 2.

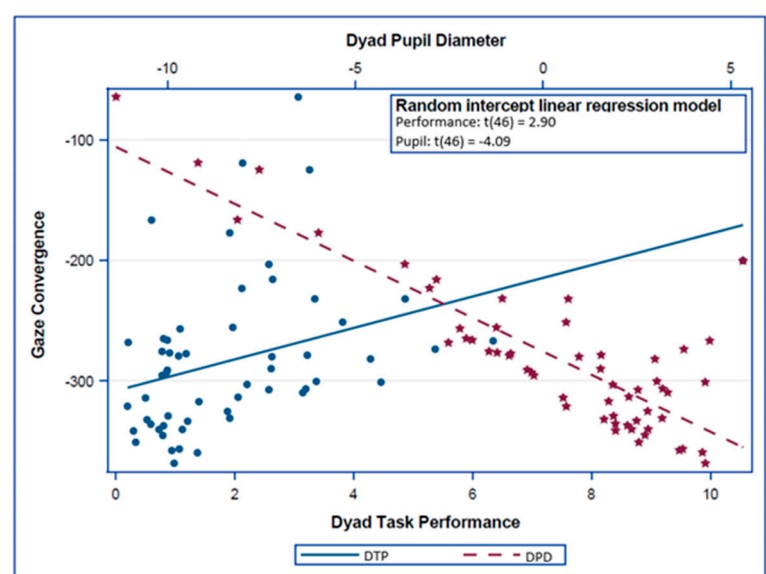

**Figure 7.** Regression plots. DTP = Dyad Task Performance; DPD = Dyad Pupil Diameter.

## 6. Discussion

The present research conducted two studies to assess (1) the content validity of dyad GC index during collaborative use of a shared system interface, and (2) predictive validity of that dyad GC index via its relationship to dyad cognitive states and performance. By addressing these points, the present study aimed to contribute to the IS and HCI literature regarding the use of dyad GC constructs in examinations of two or more users working collaboratively on a shared system interface. As mentioned earlier, studying dyad GC is important for several reasons. First, it influences group learning (e.g., Belenky et al. [2]); second, group work is impacted by the way dyads of workers look at the screen (e.g., Kwok et al. [9]); third, dyad GC helps build similar mental models within the dyad to facilitate coordination; finally, many systems designed for single user are actually used by multiple

users together, involving users gazing at visual artefacts within a system interface or at each other. Given this importance of dyad GC, our validation of the proposed dyad GC index is useful for any study relying on GC to investigate multiuser human-computer interaction. Overall, our results were very statistically significant at $\alpha = 0.001$, $\alpha = 0.01$, and $\alpha = 0.05$, giving a high degree of confidence that the investigated relationships do hold. These results are promising for future dual-eye-tracking research.

## 6.1. Content Validity

As for content validity, our proposed dyad GC index permitted clear and accurate discrimination of convergent and divergent gaze behaviors. Post-hoc analyses further confirmed this across the entire data distribution. Overall, Study 1 demonstrated that our proposed dyad GC index is in line with the conceptual definition of the dyad GC construct.

## 6.2. Predictive Validity

In line with our hypotheses, we observed that, the greater the GC of dyads, the better they performed on the Change Blindness task. As suggested in the literature, looking at the same locations reduced coordination efforts and incidents [9,34], facilitating better achievement of the dyad's task at hand [32,35,36,56]. As mentioned earlier, high performance related to answer accuracy and short trial completion time. Differences in trial completion time revealed that globally two main scenario types were observed. Dyads spending a lot of time performing the task generally ended up giving up or choosing the wrong answer. However, dyads who were fast in performing the image difference identification task generally found the right answer. In this scenario, GC was high because dyads' collaboration helped them visually converge together towards the target quickly. It may be that, for instance, as the two dyad members had to both be ready before they could move to the question, the dyad member finding the answer first and quickly was efficient in making his/her partner find it on the screen. Pehaps dyad communication influenced dyad GC, which was positively associated with higher performance. However, because our second experimental study aimed at demonstrating predictive validity of the GC index, we focus on dyad GC as an independent variable.

Also in line with our hypotheses, the greater the GC of dyads, the lesser cognitive load they exhibited. Actually, as literature suggests, during interactions between two persons, gaze direction is special in producing shift of attention on the other person [57,58]. Hence in our context, at any time during interactions within a dyad, a dyad member's gazing at a location of the screen is likely to shift the other member's gaze to the same location when the former communicates to the later where he or she intends to look at. This behavior was typical and recurrent during dyad collaboration in Study 2. When dyad members interacted during the experimental task (e.g., discussing a specific visual element in the system interface), they were thus likely to look in the same regions of the screen to indicate to each other what their actions or words refer to on the interface. It may be that, knowing that the other dyad member looked at the same location during interaction reduced the cognitive load needed to make him or her shift attention to specific regions of the shared interface. A reason may be that looking at the same regions on the screen improves harmonization of dyad members' mental models of the shared system interface [33]. Moreover, because shifting their partner's attention and gaze to screen areas required communication, faulty communication between dyad members during the task likely led to lower gaze convergence and longer times taken to look at and process the image before answering the related question. The longer they visually processed the image the more cognitive resources it took them to perform the task, that is, the higher their cognitive load. This logic applies to all trials, whether a dyad found the right answer or not, and whatever the length of time the dyad took to complete the trial. However, no association was found between the trial analysis time and cognitive load. Hence, perhaps this finding can be explained by the fact that it is more the actual visual processing of the image than the time looking at it that impacted user cognitive load. Moreover, participants were more likely to visually process the images to a greater extent when they spent more time looking at the images.

## 6.3. Advantage of Real-Time Synchronized Gaze Recording in Multiuser Human-Computer Interactions Setting

An advantage of synchronizing participants' gaze recording is that it brings order to the gaze data files and eases later data analyses, allowing the tracking of the exact time and order in which the gaze data were collected [7]. Moreover, synchronizing participants' gaze recording permitted us to develop a measure of gaze convergence using perfectly temporally aligned data, unlike existing measures that are based on Euclidian distance and use temporally asynchronous fixation data. This is a major advantage provided by our synchronized dual eye-tracking setup. Major methods for comparing two eye movement sequences based on Euclidian distance between pairs of eye fixation points [47] were proposed by Mannan et al. [49], Mathot et al. [59] and Henderson et al. [48]. Mannan et al. and Mathot et al. analyze similarity between two participants' fixation sets by computing their index of similarity based on the linear distance between each fixation point in the first fixation set and its nearest neighbor in the second fixation set, as well as the linear distance between each fixation point in the second fixation set and its nearest neighbor in the first fixation set. As the spatial distance here is always computed against the nearest neighbor in the other fixation set, a problem with this algorithm is that it does not consider the spatial variability in the distribution of the fixation sets [48]. For example, if all fixations in the first set are circumscribed within a very small area of the screen and only one fixation point in the second set is within that same area, then all fixation points in the first set will be compared to only that point from the second set. On the other hand, Henderson et al. analyze similarity between two participants' fixation sets as follows. In order to assign each fixation point in the first set to a fixation point in the second set, they analyze all possible assignments of fixation points in the first set to fixations in the second set, to find the unique assignment producing the smallest average deviation. A limitation of these algorithms is that they disregard the ordering of the fixation points of the two participants. Thus, these methods are suitable only when temporal ordering of eye fixations is not important, a condition that does not fit to the definition of the RTGD reflective measure of our SOGC construct (see Table 1). The measure of RTGD in our context of joint use of shared system interface has been facilitated by the synchronized dual eye-tracking paradigm.

Moreover, in general, despite the advantages provided by existing methods for comparing two eye movement sequences, some are subject to some issues such as not considering the temporal ordering of gaze data (Linear Distance and Edit Distance methods), data loss (MultiMatch method), high sensitivity to small temporal and spatial differences and not accounting for fixation duration (Sample-based measures) [47]. Hence, a straightforward benefit of our study is that we measure the dyad GC construct with data perfectly aligned in time and validate it in terms of content validity and predictive validity. Such measurement is appropriate to IS and HCI contexts. Since the synchronized data are not truncated nor shrunk, it accounts for the natural happening of joint use of shared system interfaces, as it is based on perfectly aligned data and acknowledges the variety of ways user dyads may look together at the screen, including all instants of diverging or converging users' gazes.

## 6.4. Contributions

The present paper theoretically contributes to the IS and HCI literature in demonstrating that GC is a valid construct with predictive validity towards performance and the cognitive states of users engaged in simultaneous collaboration on a shared user interface. Hence, this study illustrates and promotes the investigation of antecedents and consequences of GC in the IS and HCI fields. This piece or work additionally methodologically contributes to the IS and HCI literature by illustrating the feasibility of synchronous eye-tracking data recording of two or more users, a technique that improves the accuracy and simplicity of corresponding data analyses [7]. This study is one of the first ones that measure and validate gaze convergence in a synchronized multi-eye-tracker setting, involving verbal collaboration on a computerized task. Moreover, our GC index can be used in studies requiring synchronized dual eye-tracking of participants. Our experimental setup allows for real-time exploitation of GC information based on synchronized fixation data, improving accuracy of such information. For example, in addition

to or in place of the display of the other participant's gaze, our experimental setup permits the real-time display of GC information of user dyads working on the same shared interface. Implementing similar experimental setups could be used in studies examining how individuals can benefit from peers and experts, projecting real-time synchronized gaze convergence cues (e.g., D'Angelo and Begel [60]; Jarodzka et al. [61]; Król and Król [62]; Sharma et al. [63]).

Overall, the present study marks and important first step in establishing a multiuser GC index based on synchronized fixation data of user dyads and in demonstrating its utility in predicting IS and HCI constructs.

### 6.5. Implications and Research Perspectives

Dual-eye-tracking technology has been of practical utility in research. For example, some studies in multiuser settings used eye-trackers to display and share user gaze information (e.g., Nyström et al. [7]; Zhang et al. [64]). In that context, users self-assess gaze convergence based on their coworker's gaze information. The present study can contribute to investigations of impacts of real-time gaze information display in multiuser settings. For example, research works may involve enriching above gaze information by providing users working together on an interface with real-time numerical information or visual cues (e.g., color gradients) to indicate the extent to which their gazes converge at any point of time as well as for time ranges. This information may enrich collaborative use experience, permitting the examination of gaze information impacts on collaboration. For instance, just as Kwok et al. [9] found that partner user's gaze information display improved collaborative remote surgery, it may be examined whether synchronized real-time gaze convergence gradient display may improve collaborative remote surgery, which uses a shared software to remotely control robots operating surgery [9]. Moreover, as suggested in the literature, seeing a coworker's gaze information may promote users' attention shift triggering insight problem solving [53]; in addition, learning can be improved by showing an expert's information to the learner [7]. In this regard, future research may investigate the extent to which gaze convergence information may promote shifts of attention and performance during collaborative use of a shared system interface.

The present study opens multiple avenues for further research regarding gaze convergence during collaborative use of a shared interface. For instance, research could investigate possible relationships between gaze convergence and system use-related constructs, namely, emotions, cognitions and behaviors. In this respect, system use-related constructs could be investigated as consequences as well as antecedents of gaze convergence. For example, eye-tracking-based user information may be used to examine relationships between user dyads' gaze behaviors and user emotions during collaborative use of shared system interface. Such information could be exploited to improve recommendation systems' advices (for an example of a recommendation system based on user gaze and emotions, see Jaiswal et al. [65])

Another avenue for research could be to investigate team adaptation in the context of collaborative use of a shared system interface. Research could examine how IS triggers, namely, system-related, task-related, and collaboration-related triggers, impact the way IT users collectively look at a system interface, ultimately influencing performance. Moderating factors could be examined in terms of system, task, individual, group, and organizational characteristics. For instance, as literature suggests that system-related discrepant events are detrimental to performance [26], it may be investigated how such events influence gaze convergence and performance in multi-user settings.

Additionally, the success of the present study warrants further for groups of three and more users simultaneously collaborating on a shared system interface. For instance, influence of gaze convergence on learning for groups of students taking a virtual class and collaborating through the computer-supported collaborative learning system may be investigated. Likewise, it may be useful to examine how gaze convergence of groups of relatives is related to buying decisions. A tentative generalization of our operationalization of gaze convergence could involve considering the average Euclidian distance matrix of the set of users in a group, with every matrix element representing

Euclidian distance between two specific users' gaze locations on the system interface overtime. Hence, such a matrix would be made of two by two computations using the present study's proposed dyad gaze convergence index.

Furthermore, our Study 2 was done in hedonic settings, with promising findings. As our hypotheses were not specific to hedonic settings, future real-time dual eye-tracking research could be conducted in utilitarian settings to support our GC index validity as well as the generalizability of our findings to other contexts. As the scope of our study is joint task performance using a shared system interface, it is expected that in utilitarian settings, the users' looking at the same visual artefacts in the screen will be important for the same reasons presented in the hypothesis development. Looking at the same locations on the screen is expected to help harmonize users' mental model of the system and improve collaboration during the joint task. Examples of utilitarian settings include the following. Our GC index could be examined during a learning task involving a teacher (or an expert) and a learner—or learners and their peers—interacting together with a shared system interface, to better understand how gaze convergence impacts learning performance. To illustrate, existing studies use asynchronous eye movement comparison to demonstrate that people may benefit from learning what experts have looked at (e.g., Jarodzka et al. [61]; D'Angelo and Begel [60]) or that individuals perform better when they get real-time feedback on their gaze convergence vis-à-vis high-performing peers (e.g., Król and Król [62]; Sharma et al. [63]). Hence, the present study's setup—more generally, synchronized multiuser-eye-traking—could enrich these studies with the real-time gaze synchronization aspect. Moreover, during a professional task jointly performed by a dyad of workers, our GC index could be used to identify patterns of collaborative use of a shared interface that foster lower cognitive load and higher performance.

Additionally, our experimental setting—that is, joint task performance using a shared display on separate computer monitors—is only one of several common settings used for joint task performance. For example, it is common that two users sit in front of a single computer monitor to jointly perform a task. Future studies could examine gaze convergence with users sharing the same computer monitor. The model developed in this paper also applies in this context, which is very common in real business environments.

Finally, future research may focus on the measure and validation of the gyad GC construct through the OFD reflective dimension we proposed (see Figure 1), in the context of joint use of shared a system interface. Our experimental setup could be used in such purpose to collect perfectly synchronized dyad gaze data. However, the OFD dimension by definition does not require temporal ordering of gaze data; it looks for overall similarity in both dyad members' gaze behaviors and its focus is instead fixation positions and duration. Hence, to measure OFD in future research, existing methods of gaze similarity measures can be considered while benefiting the temporal precision and related spatial (fixation positions) accuracy provided by real-time synchronized dual eye-tracking.

### 6.6. Limitations

The first limitation of the present study resides in the fact that the measurement model was assessed through the SOGC dimension. Future research may focus on the operationalization of mutual gaze convergence and investigate its nomological network. For example, it may be examined how mutual gaze convergence is related to interpersonal processes during the collaborative use of a shared system interface. Secondly, we limited our study to dyads of users for simplicity of all aspects of the research. As mentioned earlier, the present work could be used to examine groups of three or more persons. Thirdly, this study does not investigate qualitative data related to dyad behaviors during the experimental task. Such data may provide more insights about mechanisms though which dyad members' gazes converge. For example, qualitative data may clarify how the decision processes during the dyad task (e.g., jointly deciding where to investigate on the screen) may have influenced convergence of dyad members' gazes. Fourthly, because of the exploratory nature of the present study, a small sample size was used. Nevertherless, as mentioned earlier, our very significant results are

promising. Finally, this study does not control for gender and dyad familiarity, that is, whether dyad members knew each other at the time of the study. Indeed, literature suggests that group familiarity may influence group collaboration and team performance (e.g., Cattani et al. [66]; Janssen et al. [67]) and that males and females may display different gaze patterns (e.g., Abdi Sargezeh [68]). Nevertheless, related risk was mitigated: participants were recruited independently from each other, and dyad were formed randomly; moreover, to analyze data related to the collaborative task, we used linear regression with random intercept model, which accounts for random effects. Notwithstanding, the role of dyad familiarity in the present context of collaborative use of a shared system interface is worth investigating in future research.

## 7. Conclusions

This paper investigates factors related to eye movements of IT user dyads during simultaneous collaboration on a shared system interface. A central concept is users' GC whose importance in the IS and HCI literature, although implied, has not been directly investigated nor measured. In the present study, we propose and test a dyad GC index. This GC index is validated in two studies. In the first study the GC index' content validity was demonstrated, as the measurement instrument was able to clearly distinguish convergence from divergence. In the second study, the GC index's predictive validity was strongly supported: as expected, we found that GC was positively associated with dyad task performance and negatively associated with dyad cognitive load. This paper has several contributions to the IS and the HCI literatures, in which research on GC's nomological network is lacking. Hence, we hope this piece of work will foster more investigations of multiuser eye-tracking in the IS and HCI fields.

**Author Contributions:** Conceptualization, A.Q.T., J.B. and P.-M.L.; methodology, A.Q.T., J.B. and P.-M.L.; validation, J.B. and P.-M.L.; formal analysis, A.Q.T.; investigation, A.Q.T., J.B. and J.A.T.; resources, P.-M.L., S.S. and M.F.; data curation, P.-M.L., S.S. and M.F.; writing—original draft preparation, A.Q.T.; writing—review and editing, P.-M.L., J.B., S.S. and M.F.; supervision, P.-M.L.; project administration, J.B.; funding acquisition, P.-M.L. All authors have read and agreed to the published version of the manuscript.

**Funding:** This research was funded by the Natural Sciences and Engineering Research Council of Canada, grant number RGPIN-2019-06602.s.

**Acknowledgments:** We are grateful for the financial support of the Natural Sciences and Engineering Research Council of Canada. We are also grateful to Shang Lin Chen and Emma Rucco for their valuable help during the present study.

**Conflicts of Interest:** The authors declare no conflict of interest. The funders had no role in the design of the study; in the collection, analyses, or interpretation of data; in the writing of the manuscript, or in the decision to publish the results.

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
