# Peer review of "Collaborative Use of a Shared System Interface: The Role of User Gaze—Gaze Convergence Index Based on Synchronous Dual-Eyetracking"

_applsci, doi:10.3390/app10134508_

Round 1

Author Response

[Please see the attachment.]

Dear reviewer,

We thank you for all your comments, which were valuable to our piece of work.Please see attached our responses and actions taken with regard to each comment.

Thanks and best regards,

Armel Quentin Tchanou

Reviewer 2 Report

This study measured the Gaze Convergence (GC) so that, as a result, an assessment of the efficiency of the work could be made. For this purpose, GC index was used as an indicator to measure SOGC and MOGC. These indicators were able to prove all three hypotheses they had set up, and clearly demonstrated the correlation between the GC index value and the diead logic load/ the relationship between GC index and task performance. There is a contribution to this study in that even if there were existing studies of the importance of Gaze Convergence, it could not be measured directly.

I have some questions about their research.
Frist of all, The number of people participating is small. I think that a small number of people in collecting and analyzing information about eye processing can reduce confidence in the results.
Second, because the gender of the people involved in this study is different, I think that the GC between different genders or between the same genders may be different. For example, it may be possible that the index values between male and male/male/female may differ.
Finally, I think that part of the user's gaze collection can be distorted in terms of the experimental environment. Since the indicators are important for GCs in the environment where real people work, I think that through experiments, bringing GCs can be set much more effectively than what they actually use. It would be nice to provide evidence that the Gaze in the experiment and the Gaze information in the real world may not be much different.

Author Response

(The authors gave the same response as above.)

Round 2

Reviewer 1 Report

see attached file

Author Response

Dear reviewer,

My author and I are grateful for the valuable comments you made on our manuscript. They brought important insights and clarifications to our work.

We are hereby submitting the revised manuscript.

Regards

Armel Quentin Tchanou

Reviewer 2 Report

I think that authors modified and updated their article well as review comments 

Author Response

Dear reviewer,

We are gratefull for your valuable comments, which significantly improved our manuscript.

Best regards,

Armel Quentin Tchanou

Round 3

Reviewer 1 Report

I believe the paper is now OK to publish. Thank you for implementing my comments.